# Quantifying the Effect of Land Use Change and Climate Variability on Green Water Resources in the Xihe River Basin, Northeast China

**Leting Lyu, Xiaorui Wang, Caizhi Sun, Tiantian Ren and Defeng Zheng ***

College of Urban and Environment, Liaoning Normal University, Dalian 116029, China;
lvleting@lnnu.edu.cn (L.L.); 18235496618@163.com (X.W.); suncaizhi@lnnu.edu.cn (C.S.);
Ren_Tiantian@163.com (T.R.)

**\*** Correspondence: defengzheng@lnnu.edu.cn; Tel.: +86-0411-84258364

**Abstract:** Based on a land use interpretation and distributed hydrological model, soil and water assessment tool (SWAT), this study simulated the hydrological cycle in Xihe River Basin in northern China. In addition, the influence of climate variability and land use change on green water resources in the basin from 1995 to 2015 was analyzed. The results show that (1) The $E_{NS}$ (Nash-Sutcliffe model efficiency coefficient) and $R^2$ (coefficient of determination) were 0.94 and 0.89, respectively, in the calibration period, and 0.89 and 0.88, respectively, in the validation period. These indicate high simulation accuracy; (2) Changes in green water flow and green water storage due to climate variability accounted for increases of 2.07 mm/a and 1.28 mm/a, respectively. The relative change rates were 0.49% and 0.9%, respectively, and the green water coefficient decreased by 1%; (3) Changes in green water flow and green water storage due to land use change accounted for increases of 69.15 mm and 48.82 mm, respectively. The relative change rates were 16.4% and 37.2%, respectively, and the green water coefficient increased by 10%; (4) Affected by both climate variability and land use change, green water resources increased by 121.3 mm and the green water coefficient increased by 9% in the Xihe River Basin. It is noteworthy that the influence of land use change was greater than that of climate variability.

**Keywords:** climate variability; land use; green water resources; Xihe River Basin

---

## 1. Introduction

The concept of green water was first proposed by Falkenmark [1] in 1995 in his study on the evaluation of the influence of water resources on crop growth. Green water resources refer to the water from precipitation that is stored in unsaturated soil and supports plant growth. It plays an important role in preserving the production and service function of land ecosystems. Green water can be classified into two components: green water flow, which refers to actual evapotranspiration, and green water storage, which refers to water stored in the soil [2]. From the perspective of the entire hydrological cycle and water balance, green water (consumed by evapotranspiration in forests, grasslands, crop lands, and wetlands) accounts for 65% of the total global precipitation, whereas blue water accounts for only 35%. Over 80% of global grain production relies on green water [3]. Unlike surface water and groundwater, green water cannot be directly extracted, transported, utilized, and managed, but can be developed and leveraged by some indirect methods. At present, organizations, including the Stockholm International Water Institute, International Fund for Agriculture Development, and Global Water System Project, have carried out researches on green water in food security. Water shortage has posed a threat to grain production in China's Loess Plateau [4]. Climate variability and land

use change have a major influence on the hydrological cycle and green water resources in the river basin. Climate variability is reflected by changes in meteorological elements including rainfall and temperature [5,6], while land use change influences green water resources in river basins through soil water holding capacity, surface evapotranspiration, and so on [7]. Therefore, studying the influence of climate variability and land use change on green water resources can provide theoretical bases for regional water resource management and ecological water resource planning.

Currently, three methods are mainly used for evaluating green water: biological, hydrological, and bio-hydrological approaches [8]. The hydrological modeling method uses models to estimate the amount of green water resources at the river basin scale based on the theory of water balance. This method reveals the in-depth mechanism of hydrological processes at low cost, with easy access to research data. Thus, this approach has attracted extensive attention worldwide. Common hydrological models include relatively large-scale models, such as Hydrological Land Use Change (HYLUC) and STREAM, and small-scale models, such as Agricultural Catchments Research Unit (ACRU) and Soil and Water Assessment Tool (SWAT). The SWAT model also has been widely applied [9]. Faramarzi et al. [10] used SWAT in combination with the Sequential Uncertainty Fitting (SUFI-2) at the sub-basin level, and discussed the spatial and temporal distribution of blue and green water resources in Iran. They found that irrigation methods had a major influence on hydrological cycles. Schuol [11] calibrated and validated the SWAT model at 207 hydrological stations across Africa, evaluating the spatial and temporal distribution of available blue and green water resources. Xu et al. [12] reviewed the concepts and evaluation methods of blue and green water, harnessed the SWAT model combined with SUFI-2, and assessed the spatial and temporal distribution of blue water, green water flow, and storage resources in the Weihe River Basin in the last 50 years at multiple scales. This research provided the scientific basis for water resource planning and management in northwestern China. Zhao et al. [13] leveraged the SWAT model to analyze the spatial and temporal difference of blue and green water resources in the Weihe River basin in typical years. Zhu et al. [14] using the SWAT model simulated the HRB, putting forward effective measures for the management of green water resources. On this basis, they studied the effects of climate variability and human activities in the past 30 years on blue and green water resources in the river basin and pointed out that human activities led to changes in blue-green water resources. Overall, research on blue and green water resources is popular worldwide and is attracting increasingly more attention.

Taking the Xihe River Basin as an example, this study applied the SWAT model and simulated the hydrological cycle under three scenarios to quantify the effect of climate variability and land use change on the amount, and spatial and temporal distributions of green water resources. The research results can provide a scientific basis for planning and management of regional water resources.

## 2. Material and Methods

### 2.1. Research Area Overview

The Xihe River Basin is located in Liaoning Province, China, between 40°45′ N–41°00′ N and 123°40′ E–124°00′ E. It is a tributary of Taizi River, with a length of 200.737 km, and a drainage area of 1111 km$^2$. The Xihe River basin has a typical temperate monsoon climate, characterized by hot, rainy summers, and cold, dry winters. The terrain of the river basin is higher in the east and south, and lower in the west and north. The downstream mainly lies in plain areas (Figure 1).

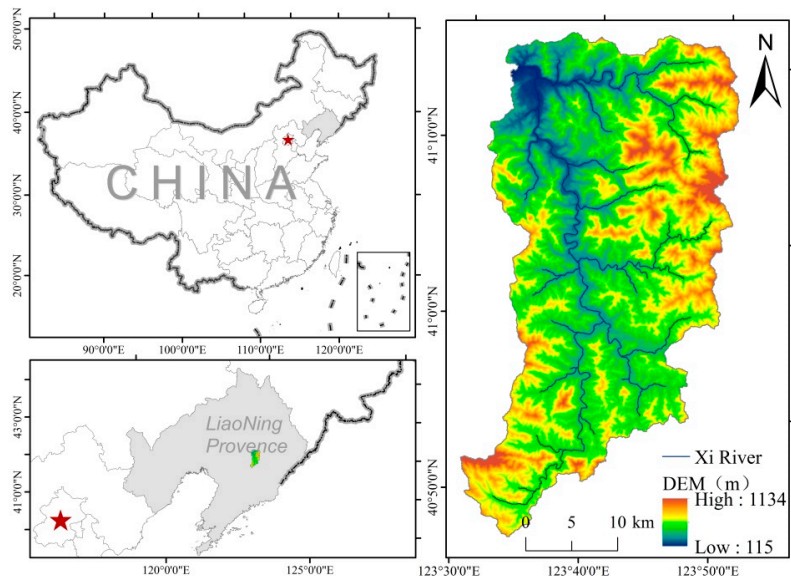

**Figure 1.** Location and general characteristics of Xihe River Basin.

*2.2. Data Sources*

In this study, digital elevation model (DEM) data with a 30 m × 30 m spatial resolution were obtained from international data service platforms. Soil data (scale of 1:100,000) were acquired from the Harmonized World Soil Database. By reviewing the statistical yearbook, the daily precipitation records from 1975 to 2015 were obtained, which were provided by precipitation stations in Gujiatun, Nanfen, Qiaotou, Xiamatang, Yongziyu, and Qiaotou in the Xihe River Basin. Meteorological data, such as daily precipitation, temperature, solar radiation, wind speed, and relative humidity provided by Benxi meteorological station, for the study period (1975–2015) were retrieved from the official website of the China Meteorological Data Service Center. Daily runoff records in Qiaotou hydrologic station for the study period was also obtained for model calibration (Table 1).

**Table 1.** Information of meteorological and hydrological stations in the River Basin.

| Station | Data | Longitude | Latitude |
|---|---|---|---|
| Gujiatun | daily precipitation | 124°16′20″ | 41°37′35″ |
| Nanfen | daily precipitation | 124°15′18″ | 41°09′00″ |
| Xiamatang | daily precipitation | 124°13′33″ | 41°03′33″ |
| Yongziyu | daily precipitation | 124°30′15″ | 41°15′60″ |
| Qiaotou | daily precipitation and runoff | 124°10′10″ | 41°22′26″ |
| Benxi | daily precipitation, temperature, solar radiation, wind speed, and relative humidity | 124°43′06″ | 42°39′41″ |

Remote sensing images, including Landsat TM in 1995 and Landsat OLI in 2015, with a spatial resolution of 30 m were obtained from the Geospatial Data Cloud (http://www.gscloud.cn). The orbit number/line number ratios of Landsat TM and Landsat OLI were 119/31, and 119/32, respectively. For each period, two images were selected. The imaging time was between June and October, which represents the growing season with lush vegetation. The cloud cover of the images was less than 2%. Radiometric calibration, atmosphere correction, and image cutting and splicing were performed with ENVI5.1 program. According to the current land use classification standard, the study area was divided into six land use types, including crop, forest, grassland, urban and built-up, water, and unused land by the maximum-likelihood method. Using Google Earth, a confusion matrix was built to assess the accuracy. The results show that, the Kappa coefficients of the images, selected from the two periods, were both over 80% (Table 2), which meet the requirement of medium-resolution remote sensing images (Figure 2).

**Table 2.** Accuracy evaluation of land use classification of Xihe River Basin in 1995 and 2015.

| Year | Class | Number of Reference Pixels | Number of Pixels Classified | Number of Pixels Classified Correctly | Product Accuracy (%) | User Accuracy (%) |
|---|---|---|---|---|---|---|
| | Water | 168 | 171 | 167 | 99.40 | 97.66 |
| | Urban and built-up | 98 | 94 | 93 | 94.90 | 98.94 |
| | Forest | 357 | 375 | 357 | 100.00 | 95.20 |
| 1995 | Unuse | 52 | 55 | 46 | 88.46 | 83.64 |
| | Grass land | 46 | 30 | 30 | 65.22 | 100.00 |
| | Crop land | 272 | 268 | 263 | 96.69 | 98.13 |
| | Total | 993 | 993 | 956 | | |
| | | **Overall Accuracy = 96.27%** | | **Kappa Coefficient = 0.9502** | | |
| | Water | 61 | 58 | 57 | 93.44 | 98.28 |
| | Urban and built-up | 131 | 146 | 112 | 85.50 | 76.71 |
| | Forest | 291 | 277 | 274 | 94.16 | 98.92 |
| 2015 | Unuse | 104 | 134 | 102 | 98.08 | 76.12 |
| | Grass land | 103 | 62 | 49 | 47.57 | 79.03 |
| | Crop land | 141 | 154 | 141 | 100.00 | 91.56 |
| | Total | 831 | 831 | 735 | | |
| | | **Overall Accuracy = 88.45%** | | **Kappa Coefficient = 0.8537** | | |

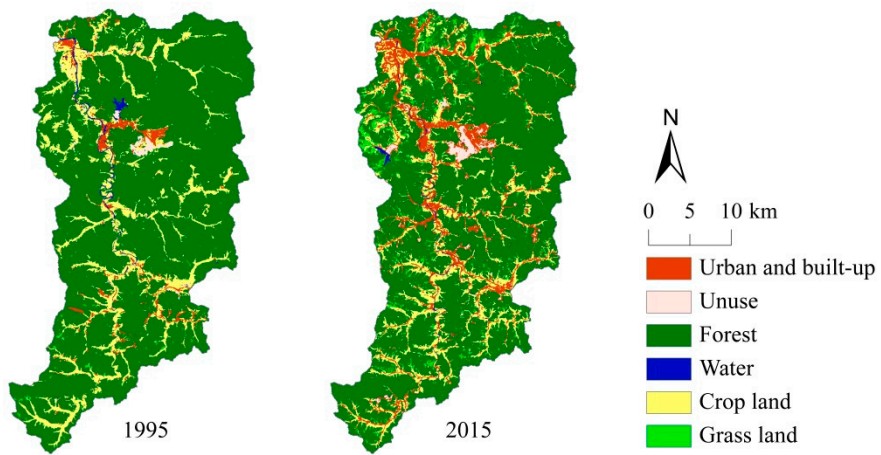

**Figure 2.** The land use maps of Xihe River Basin in 1995 and 2015.

*2.3. Methodology and Process*

2.3.1. SWAT Model

SWAT is a river basin scale model developed by the Agricultural Research Service, United States Department of Agriculture. It can simulate the quality and quantity of surface and groundwater to predict the influence of land management practices on the hydrology, sediment, and agrochemicals in large-scale river basins with various soil types, land use types, and management conditions. In SWAT modeling, the river basin is divided into many sub-basins and then each sub-basin can be further divided into several hydrologic response units (HRUs) [15,16]. The SWAT model simulates the hydrological cycle of the river basin according to the water balance equation.

$$SW_t = SW_0 + \sum_{i=1}^{t}(R_{day} + Q_{surf} - E_a - w_{seep} - Q_{gw}) \tag{1}$$

where $SW_t$ is the final soil water content, $SW_0$ is the initial soil water content on day *i*, *t* is the time, $R_{day}$ is the amount of precipitation on day *i*, $Q_{surf}$ is the amount of surface runoff on day *i*, $E_a$ is the amount of evapotranspiration on day *i*, $W_{seep}$ is the amount of seepage and bypass water from the soil profile on day *i*, and $Q_{gw}$ is the amount of return flow on day *i*.

The Nash-Sutcliffe model efficiency coefficient ($E_{NS}$) and coefficient of determination ($R^2$) are used to evaluate the consistency and credibility between the simulated and observed values. The simulation results are usually considered to be reliable when the values of $R^2$ and $E_{NS}$ exceed 0.6 and 0.5, respectively [17–19].

2.3.2. Water Resource Statistics

According to the concept of green water and output results of the SWAT model, green water flow is actual evapotranspiration (*ET*) and green water storage is soil water (*SW*). Thus, the total amount of green water resources equals the sum of actual evapotranspiration and soil water. Additionally, in this study, the green water coefficient (*GWC*) is adopted to evaluate the distribution of green water resources in the river basin. The *GWC* is the proportion of green water resources in the total water resources (green and blue water resources) [20]. Blue water resources include surface water, groundwater, and water in rivers, lakes, and aquifers. The amount of blue water can be represented by the sum of water yield in sub-basins (*WYLD*), and deep aquifer recharge (*DA_RCHG*) [3,9,21,22]. The formulas of green water resources and *GWC* are expressed as follows:

$$G = ET + SW \tag{2}$$

$$B = WYLD + DA\_RCHG \tag{3}$$

$$GWC = \frac{G}{G + B} \times 100\% \tag{4}$$

where *G* is the amount of green water resources, including green water flow and green water storage, *B* is the amount of blue water resources, and *GWC* is green water coefficient.

The relative change rate (*RCR*) of water resources in different periods can be expressed as:

$$RCR = \frac{(V_i - V_0)}{V_0} \times 100\% \tag{5}$$

where *V* is the variable, representing green water flow, green water storage, and *GWC* in this study, and *i* and *0* are variable values in the *i* period and the initial period, respectively.

### 2.3.3. Design of Simulation Scenarios

On the basis of the SWAT model, the one-factor-at-a-time method, in which one factor is fixed and the other is changed [23–27], was applied to quantitatively distinguish the effect of climate variability and human activities on hydrological cycles. The simulation scenarios [13,28] are shown in Table 3. In this paper, land use data in 1995 represent the land use status from 1990 to 1999, and land use data in 2015 stand for the land use status from 2005 to 2015. Taking Scenario I as the basis, the influence of climate variability on green water resources was obtained by comparing Scenarios I and II. The influence of land use change on green water resources was obtained by comparing Scenarios III and II, and the combined influence of climate variability and land use change was obtained by comparing Scenarios III and I.

**Table 3.** Scenario Design.

| Scenario | Land Use | Meteorological Data |
| --- | --- | --- |
| **Scenario I** | 1995 | 1990–1999 |
| **Scenario II** | 1995 | 2006–2015 |
| **Scenario III** | 2015 | 2006–2015 |

### 2.3.4. Simulation

The DEM, river system, soil type, and land use data were leveraged to establish the spatial database for the SWAT model. The attribute database included land use and soil type attribute data, and methodological data (Figure 3). The Xihe River Basin was divided into 27 sub-basins and 1148 HRUs. Based on the SUFI-2 algorithm of SWAT-CUP, the model parameters were calibrated with monthly runoff data provided by the Qiaotou hydrological station, which is located at the river basin mouth. These calibrated parameters were validated with monthly runoff data for 1995–2015. The warm-up periods for the calibration and validation periods were 2 years each.

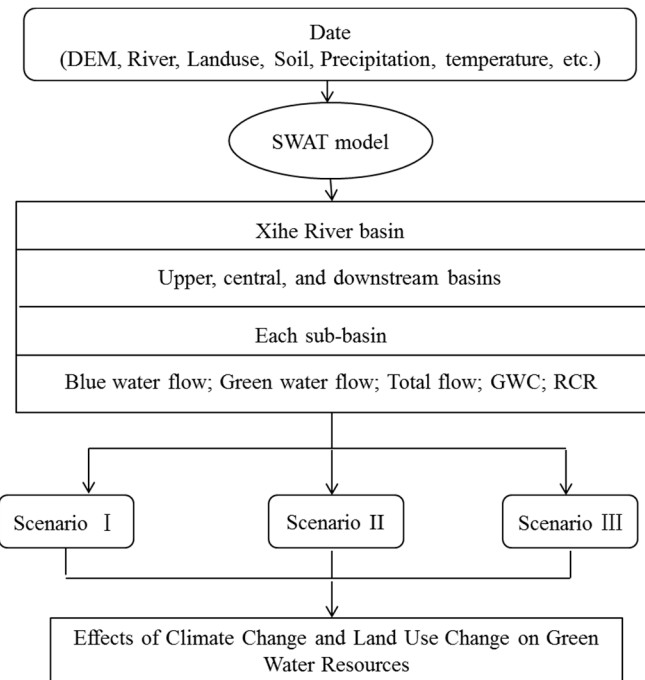

**Figure 3.** The framework for the article.

## 3. Results and Analysis

### 3.1. Analysis of Land Use Change

Land use distributions of the Xihe River Basin during various periods are displayed in Table 4. In 1995, the areas of forestland and cropland were 913.35 km$^2$ and 147.01 km$^2$, accounting for 82.17% and 13.23% of the total area of the river basin, respectively. The proportions of urban and built-up land, water, unused land, and grassland were relatively low at 2.7%, 0.8%, 0.8%, and 0.3%, respectively. In 2015, the areas of forestland, cropland, and water decreased, while those of urban and built-up land, unused land, and grassland increased. The area of forestland showed the most change, with a reduction of 106.53 km$^2$ and a change rate of −9.58%. Urban and built-up land exhibited the second largest change with an increase of 73.05 km$^2$ and a change rate of 6.57%. Changes in the areas of the other four types of land use were relatively small.

**Table 4.** Comparison of land use distributions during various periods in the Xihe River Basin.

| Periods | Land Use Change | Forest | Cropland | Urban and Built-up | Water | Unused | Grassland |
|---|---|---|---|---|---|---|---|
| 1995 | Area (km$^2$) | 913.35 | 147.01 | 30.06 | 8.89 | 8.88 | 3.28 |
|  | Percent (%) | 82.17 | 13.23 | 2.70 | 0.80 | 0.80 | 0.30 |
| 2015 | Area (km$^2$) | 806.82 | 144.26 | 103.11 | 5.43 | 16.60 | 35.25 |
|  | Percent (%) | 72.59 | 12.98 | 9.28 | 0.49 | 1.49 | 3.17 |
| 1995–2015 | Change area (km$^2$) | −106.53 | −2.75 | 73.05 | −3.46 | 7.72 | 31.97 |
|  | Change percent (%) | −9.58 | −0.25 | 6.57 | −0.31 | 0.69 | 2.88 |

### 3.2. Simulation Results of SWAT

In the model calibration period, the ENS was 0.94, and R$^2$ was 0.89; in the validation period, the E$_{NS}$ was 0.89, and the R$^2$ was 0.88 (Figure 4). The SWAT model performed well for simulation with high accuracy [17,18], and it could describe the hydrological cycle of the Xihe River Basin.

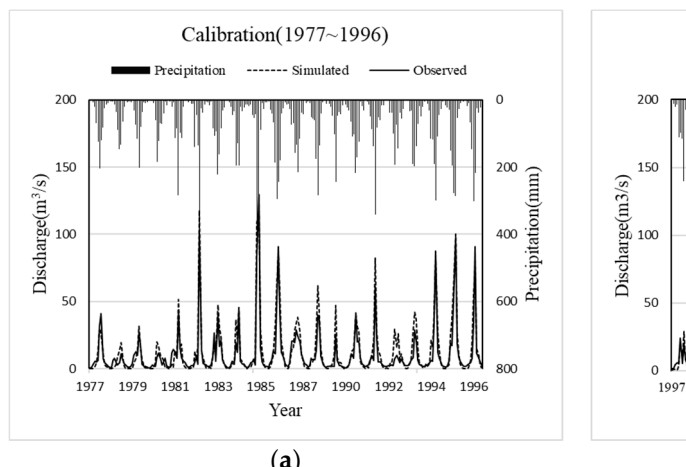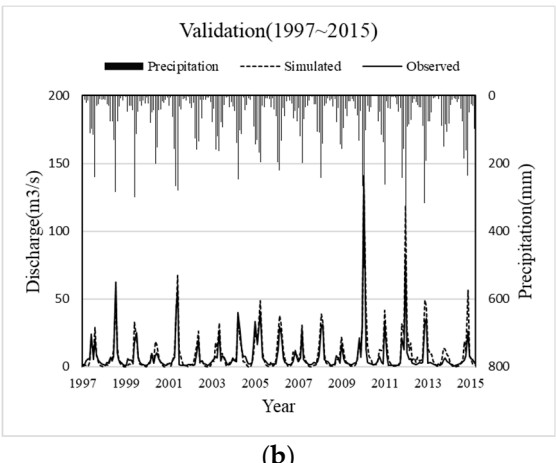

|   |   |
|:-:|:-:|
| (**a**) | (**b**) |

**Figure 4.** Comparison of (**a**) simulated and (**b**) observed values of monthly runoff in calibration and validation periods.

### 3.3. Effects of Climate Variability and Land Use Change on Green Water Resources

3.3.1. Changes of Green Water Flow

The simulation results show that the amounts of green water resources in Scenarios I, II, and III were 419.30 mm, 421.37 mm, and 490.51 mm, respectively. Under the combined influence of climate variability and land use change (Scenario III–Scenario I), the amount of green water flow increased by 71.21 mm/a. In particular, the amount of green water flow increased by 2.07 mm/a due to climate variability (Scenario II–Scenario I), and by 69.15 mm/a due to land use change (Scenario III–Scenario II).

The spatial distribution of green water flow in the three scenarios is presented in Figure 5. As shown, green water flows were lower in the middle part and higher in the peripheral parts of the river basin. According to the spatial distribution of the relative change rate of green water flow, the influence of climate variability was not obvious because the relative change rates were all under 5%, and the amount of green water flow decreased slightly in small parts in the southeast and west (change rates < 0). The effect of land use change on green water flow was greater, as shown by the change rate of above 16% in the west and south parts, and even over 20% in some parts. Under the combined influence of climate variability and land use change, the change rate was over 16% in most parts, and slightly lower in small parts in the northeast between 11% and 15%.

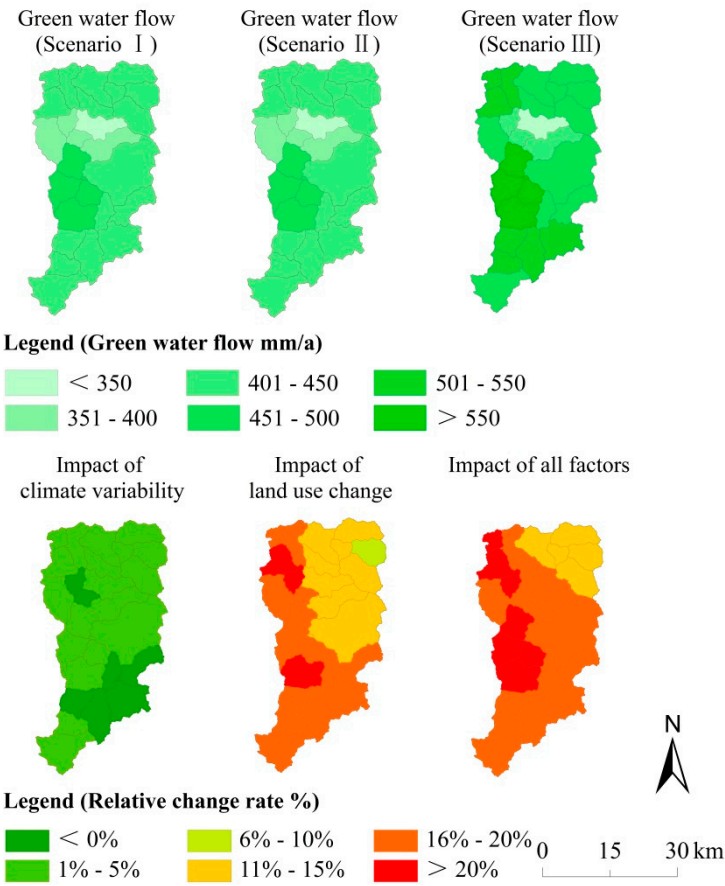

**Figure 5.** Relative change rates and spatial distribution of green water flow in the Xihe River Basin in the three scenarios.

### 3.3.2. Changes of Green Water Storage

The simulation results show that the amounts of green water storage in Scenarios I, II, and III were 129.94 mm, 131.21 mm, and 180.03 mm, respectively. Under the combined influence of climate variability and land use change (Scenario III–Scenario I), the amount of green water increased by 50.1 mm/a in total, of which 1.28 mm/a could be attributed to climate variability (Scenario II–Scenario I) and 48.82 mm/a could be attributed to land use change (Scenario III–Scenario II).

The spatial distribution of green water storage in the three scenarios is presented in Figure 6. In the three scenarios, the largest amount of green water storage was in the southeastern part of the river basin, and the smallest amount was in the middle part. With the land use data for 2015 and meteorological data for 2006–2015, Scenario III had the largest amount of green water storage, and the amount of green water storage was above 160 mm/a in most areas. In Scenarios I and II, the amount of green water storage was somewhat lower, ranging from 120 mm/a to 150 mm/a.

Climate variability had a smaller influence on green water storage with a change rate of less than 15% for the entire area, and even less than 0% for the northeastern part. In contrast, land use change had a greater influence on green water storage with a change rate of more than 31% in most parts, and over 60% in some parts. Under the combined influence of climate variability and land use change, the change rate of green water was above 31% in most parts, over 46% in the middle part, and over 60% in some parts. The change rate in the western and southwestern parts was relatively low, ranging from 16% to 30%.

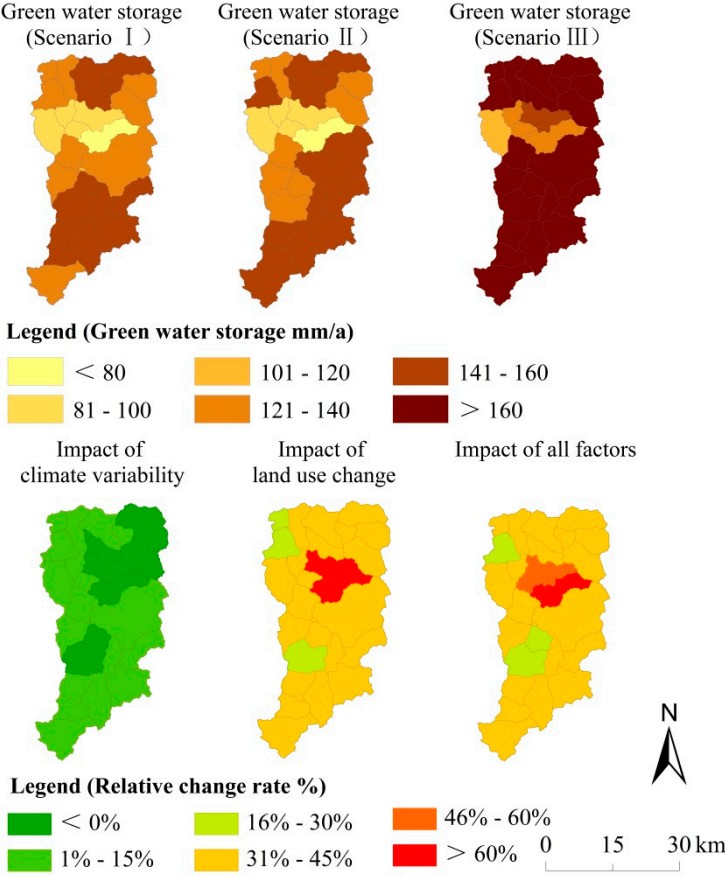

**Figure 6.** Spatial distribution and relative change rates of green water storage in the Xihe River Basin in the three scenarios.

### 3.3.3. Changes of Green Water Coefficient

The simulation results show that the green water coefficients in Scenario I, II, and III were 59%, 58%, and 68%, respectively. The green water coefficient increased by 9% under the combined influence of climate variability and land use change (Scenario III–Scenario I). Climate variability (Scenario II–Scenario I) accounted for a decrease of 1% in the green water coefficient. On the contrary, land use change (Scenario III–Scenario I) accounted for an increase of 10%.

From the perspective of spatial distribution (Figure 7), the green water coefficient was higher in the western part of the river basin and lower in the eastern part. Climate variability had a smaller effect on green water coefficient with a change rate lower than 5% for the entire area, and less than 0% in most parts, and the green water coefficient declined in most parts. Land use change had a greater effect on green water coefficient with a change rate of over 5% in general and more than 20% in some parts. Under the combined influence of climate variability and land use change, the change rate was above 17% in most parts, and slightly lower in some parts in the northeast and west, between 6% and 16%.

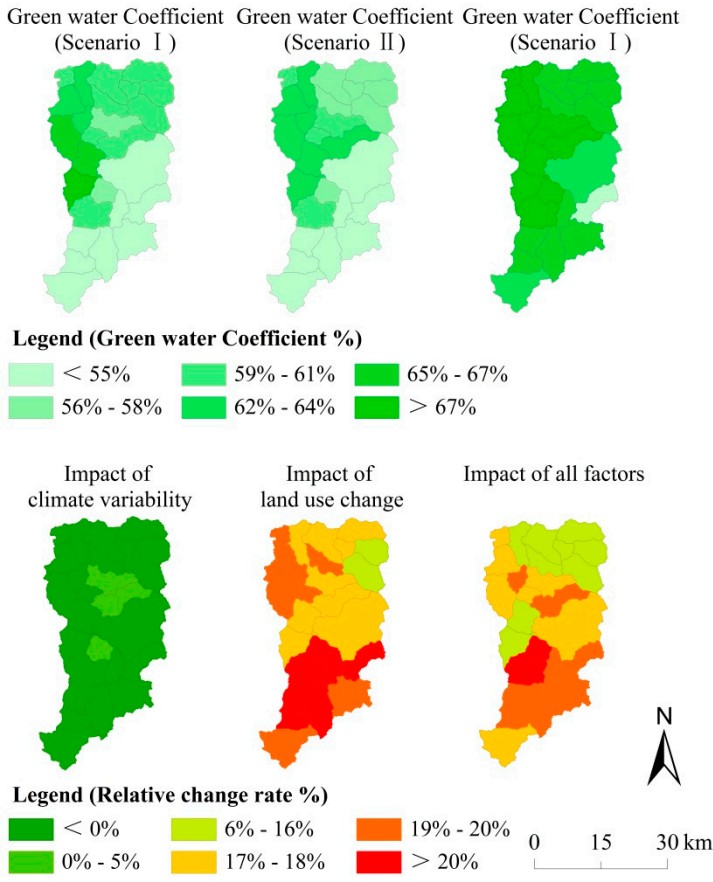

**Figure 7.** Spatial distribution of green water coefficient in the three scenarios.

## 4. Discussion

The results of this study indicate that the increase of green water resources in the Xihe River Basin is more strongly correlated to land use change than climate variability. This is different from Zhao's result of Weihe River Basin, northwest China [12]. In the Weihe River Basin, climate variability plays a major role in impacting green and blue water resources. This is because the climate has undergone drastic changes from 1985 to 2008 in this basin. Annual temperature exhibits a significantly increasing trend and precipitation recorded shows a significant decreasing trend. The climate was getting drier and warmer. Meanwhile, the land use changed slightly from 1985 to 2008 in the Weihe River Basin. Farmland has changed the most by only −1.4%. On the contrary, the land use change in the Xihe River Basin is much more dramatic (Table 4). And under the temperate monsoon climate, changes in precipitation and temperature have been relatively small during these two periods (1990–1999 and 2006–2015). Thus, land use change is the main cause of influencing the distribution of water resources in the Xihe River Basin. Similar results are also found in two Slovenian Mediterranean catchments, where the land use change is also relatively large during a longer time series [29]. In the Xihe River Basin, the increase in grassland area has changed soil water storage and caused more evapotranspiration. More precipitation was converted to green water flow, which increased by 69.15 mm/a. This plays an important role in vegetation restoration, and water and soil conservation. With increased forest area and decreased cropland area in the northern and southern parts of the river basin, the area of agricultural irrigation increased, raising the possibility of crop evapotranspiration [30–33]. Subsequently, the total volume of green water resources increased.

Both climate variability and land use change affected the green water coefficient. The spatial difference of green water coefficient was related to the rainfall distribution. With relatively high rainfall [34], the eastern part of the river basin witnessed the growth of urban and built-up land

and crop land in the last 20 years. In urban construction, the ground becomes hardened, reducing surface infiltration and flow resistance. Thus, surface runoff is increased, and the flow generation rate is elevated [35,36]. Meanwhile, surface evapotranspiration is reduced directly, and the effective evapotranspiration time is decreased. Moreover, the green water coefficient was relatively small [37] because actual evapotranspiration (green water flow) and soil water content decreased. The western part mainly consists of forestland, cropland, and grassland, with a fairly good ecological environment. After being absorbed by the soil, rainfall is converted into green water storage and green water flow, through which water is evaporated into the atmosphere. Therefore, the amount of green water resource and green water coefficient were relatively high in this region.

## 5. Conclusions

Based on land use change and the SWAT model, this study simulated the hydrological cycle in the Xihe River basin and quantified the effects of land use change and climate variability during 1995–2015. The following conclusions can be drawn:

(1) $E_{NS}$ and $R^2$ were 0.94 and 0.89, respectively, in the calibration period, and 0.89 and 0.88, respectively, in the validation period. The model showed good performance in the simulation and could be used to describe the hydrological cycle in the river basin with high accuracy.

(2) Due to climate variability, the amount of green water flow and green water storage increased by 2.07 mm/a and 1.28 mm/a, respectively, and the green water coefficient was decreased by 1%. The change rate of green water flow was below 5% for the entire area, and lower than 0% in some small parts in the southeast and west. The relative change rate of the green water storage was less than 15% in general, and below 0% in some small parts in the northeast and southwest.

(3) Affected by land use change, the amount of green water flow and green water storage increased by 69.15 mm/a and 48.82 mm/a, respectively, and the green water coefficient was increased by 10%. Green water flow was significantly affected in the western and southern parts, but it was not much affected in small parts towards the east. Green water storage exhibited the greatest change in the central part of the river basin, but the change rate was relatively low in some small parts in the northwest and southwest. In most parts of the river basin, the change rate of the green water coefficient was over 17%.

(4) Under the combined influence of climate variability and land use change, the amount of green water resources in the Xihe River basin increased by 121.3 mm, and the green water coefficient was 9% higher. Overall, land use change was mainly responsible for changes in the green water resources in the Xihe River Basin.

**Author Contributions:** Conceptualization, L.L. and D.Z.; Methodology, L.L. and D.Z.; Software, X.W. and L.L.; Validation, L.L.; Formal Analysis, X.W.; Investigation, X.W.; Resources, L.L.; Data Curation, X.W. and T.R.; Writing-Original Draft Preparation, X.W.; Writing-Review & Editing, L.L.; Visualization, C.S. and D.Z.; Supervision, C.S. and D.Z.; Project Administration, L.L.; Funding Acquisition, L.L.

**Funding:** This research was funded by the National Natural Science Foundation of China grant number 41701208.

**Conflicts of Interest:** The authors declare no conflict of interest.

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
