# Peer review of "Quantifying the Effect of Land Use Change and Climate Variability on Green Water Resources in the Xihe River Basin, Northeast China"

_sustainability, doi:10.3390/su11020338_

Reviewer 1 Report

General      Overview

Authors of this manuscript have presented a study related to the effect of land use change and climate change on green water resources in the xihe river basin, northeast China. They have simulated the hydrological cycle in Xihe River Basin in northern China Based on a land use interpretation and distributed hydrological model, Soil and Water Assessment Tool (SWAT). They have found that the influence of land use change was higher than that of climate change. The topic is interesting, an important issue and generally well written and structured. However, there are still some occasional grammar errors through the manuscript especially the article ‘’the’’, ‘’a’’ and ‘’an’’ is missing in many places, please make a spellchecking.

The results and discussion section needs further improvement, compare your findings with the other author's conclusions. Please provide more deep discussion about your results, compare your findings with the other author findings. Please clearly state the novelty of this work. Please check the reference style, some of the references are not according to the journal style, especially the journals abbreviations. Therefore, the reviewer recommends to further improve the manuscript before accepting it for publication. Some of the specific comments are listed below.

Specific      Comments   

The reviewer recommends renaming section 2 into Material and Methods.

In many places’  articles ‘’the’’, ‘’a’’, ‘’an’’ is missing, please check and correct.

It will be excellent if the authors could explain the methodology through the flowchart but in a more symbolic way, not just text.

What climate model did the authors used, this is not clear and explicitly explained?

Figure 1, has low resolution, please improve it.

Please explain why green water flows were lower in the middle part and higher in the peripheral regions of the river basin?

Did the authors use soil moisture data, if yes at what resolution?

Please provide deeper discussing by comparing your findings with the literature.

There is a lack of references; please consider citing the following literature:

Wang, J.; Ma, J.; Clarke-Sather, A.; Qu, J. Estimating Changes in the Green Water Productivity of Cropping Systems in Northern Shaanxi Province in China’s Loess Plateau. Water 2018, 10, 1198.

Kuriqi, A. (2016). Assessment and quantification of meteorological data for implementation of weather radar in mountainous regions. MAUSAM, 67(4), 789-802.

Matohlang Mohlotsane, P.; Owusu-Sekyere, E.; Jordaan, H.; Barnard, J.H.; van Rensburg, L.D. Water Footprint Accounting Along the Wheat-Bread Value Chain: Implications for Sustainable and Productive Water Use Benchmarks. Water 201810, 1167.

Zhu, K.; Xie, Z.; Zhao, Y.; Lu, F.; Song, X.; Li, L.; Song, X. The Assessment of Green Water Based on the SWAT Model: A Case Study in the Hai River Basin, China. Water 201810, 798.

Please check the reference style, some of the references are not      according to the journal style, especially   the journals abbreviations.

Concluding Remarks

The work presented in this manuscript is an interesting topic, it needs some more efforts to improve it further. Reviewer recommend minor revision of this manuscript and publishing it only after specific improvement of the current version are made.

Author Response

Dear Reviewer:

We are so sorry that there are grammar errors in our manuscript entitled ‘sustainability-404523’ and thank you for your comments. These comments are all valuable and very helpful for revising and improving our paper. The main modification are as flowing:

1、We have increase the discussion about our results.

2、We have changed the reference style in our manuscript.

3、We have explain our methodology through the flowchart.

4、We didn’t use any climate model. Daily meteorological data we used were measured data provided by China Meteorological Data Service Center (http://dataNaNa.cn/en). And we use the weather generator provided by SWAT model to interpolate the missing data.

5、We have improves the resolution of Figure 1.

6、Green water can be classified into two components: green water flow, which refers to actual evapotranspiration, and green water storage, which refers to water stored in the soil. The landuse type of peripheral regions is mainly forest. And trees have the function of water conservation, which means green water storage in this area are relatively higher. And in the middle part of the basin, crop land, urban and built-up is the main landuse type. More water resources evaporate into the air. So green water flows are lower in this area are higher compared with the peripheral regions.

7、We didn’t use soil moisture data. We use the Harmonized World Soil Database, from HWSD(Harmonized World Soil Database, HWSD http://westdc.westgis.ac.cn/). Including SNAM, NLAYERS, HYDGRP, SOL_ZMX, ANION_EXCL, SOL_CRK, TEXTURE, SOL_Z, SOL_BD, SOL_AWC, SOL_K, SOL_CBN, CLAY, SILT, SAND, ROCK, SOL_ALB, USLE_K, SOL_EC. There SOL_BD, SOL_AWC are related with soil moisture.

8、We have deeper discussing by comparing your findings with the literature.

9、We have added the references that you suggested.

Reviewer 2 Report

The article entitled “Quantifying the Effect of Land Use Change and Climate Change on Green Water Resources in the Xihe River Basin, Northeast China”, seeks to provide a comparison between three different periods using the Soil and Water Assessment Tool (SWAT). First of all, I would like to noted that, scenarios provide a mechanism for the forecasting, thus it is a tool of long-range outlook and it is related with various approaches including the water resources. In current effort, authors used the term for three deferent past periods as portrayed in table 1. However, we can say that scenario is defined as a or more future history / ies. I suggest to change the term or please justify in the introduction the term selection. Additional, the authors are explaining the changes in these periods are strongly correlated to land use change than climate change. I think this a general comment and it is essential to identify the drought events in these periods from the precipitation data in the case study area. The objective of defining green water flow has an uncertainty, also, I would like to remind the following terms for the types of evapotranspiration “potential”, “reference” and “actual” is to eliminate the crop specific changes in the evapotranspiration process. In the "potential" evapotranspiration definition, this is attempted by assuming the constant crop conditions.  However, in such a definition, the crop is not very well specified, and this may create a problem in the total elimination of crop component. [1] More specifically, the potential evapotranspiration (ETp) concept was first introduced in the late 1940s and 50s by Penman and it is defined as “the amount of water transpired in a given time by “a short green crop, completely shading the ground, of uniform height and with adequate water status in the soil profile”. Note that in the definition of potential evapotranspiration, the evapotranspiration rate is not related to a specific crop [maximum rate of ET). Scientists may be confused as to which crop to select to be used as a short green crop, because the evapotranspiration rates from well-watered agricultural crops may be as much as 10 to 30% greater than that occurring from short green grass. The reference evapotranspiration (ETo) concept was introduced by irrigation engineers and researchers in the late 1970s and early 80s to avoid ambiguities that existed in the definition of potential evapotranspiration. So, Reference evapotranspiration (ETo) is defined as: "the rate of evapotranspiration from a hypothetical reference crop with an assumed crop height of 12 cm , a fixed surface resistance of 70 sec/m and an albedo of 0.23, closely resembling the evapotranspiration from an extensive surface of green grass of uniform height, actively growing, well-watered, and completely shading the ground [2,3,4]. In the reference evapotranspiration definition, the grass is specifically defined as the reference crop and this crop is assumed to be free of water stress and diseases. Two main crops have been used as the reference crop, grass and alfalfa. Furthermore, the weather data collection site is well defined in the ETo definition and the irrigated grass area of the weather data collection site should be fairly large. Last but not least, the Actual evapotranspiration [2,6,9] denoted as ETa, is the amount of water that is actually removed from a surface due to the processes of evaporation and transpiration, under the (current) condition of existing water supply or actual evapotranspiration (AE) is the amount of water that evaporates from the surface and is transpired by plants if the total amount of water is limited.[8]. Actual crop ET can be measured directly or indirectly (only at a plot scale) in fields by methods as Soil water balance, weighing lysimeter, energy balance/Bowen ratio, aerodynamic method, eddy covariance, sap flow method, etc). Most of such data at a plot scale required for these methods are not available in the Case study area, as well as in most European countries. Finally, please insert tables with the list of hydrological and meteorological stations including coordinates, elevation, and descriptive statistics of the climatic parameters.

Please consider the above comments in the current methodology.

References

[1] Suat Irmak and Dorota Z. Haman - IFAS Extension, ABE, 2003

[2] Allen R.G., Pereira L.S., Raes D., Smith M. (1999) Crop Evapotranspiration. Guidelines for Computing Crop Water Requirements. FAO Irrigation and Drainage Paper No. 56. Rome, Italy: United Nations – FAO. 300 pp.

[3] Doorenbos, J. and Pruitt, W.O. (1977) Crop Water Requirements. FAO Irrigation and Drainage Paper 24, FAO, Rome, 144 p.

[4] Pruitt, W. O. (1966), Empirical method of estimating evapotranspiration using primarily evaporation pans, in Evapotranspiration and Its Role in Water Resources Management, pp. 57–61, Am. Soc. of Agric. and Biol. Eng., St. Joseph, Mich.,

[5] Rana, G.; Katerji, N. Measurement and estimation of actual evapotranspiration in the field under Mediterranean climate: a review. European Journal of Agronomy 2000, 13, 125–153.

[6] Jensen, Marvin E., Ph.D., NAE; and Richard G. Allen, Ph.D., P.E (Editors), 2016. Evaporation, evapotranspiration, and irrigation water requirements, 2016. Task Committee on Revision of Manual 70

 [7] GH Hargreaves, ZA Samani, (1985). Reference crop evapotranspiration from temperature.  Applied engineering in agriculture, elibrary.asabe.org

[8] A Dictionary of Earth Sciences third edition Edited by Michael Allaby. Oxford University press

[9] Pidwirny, M. (2006) The Drainage Basin Concept. Fundamentals of Physical Geography, 2nd Edition.

Author Response

Dear Reviewer:

Thank you very much for your teach about potential evapotranspiration, reference evapotranspiration and actual evapotranspiration. It’s so clearly and patiently, we have learned a lot. And we have add a table about information of hydrological and meteorological stations according to your advice.

Reviewer 3 Report

This paper is focused on the use of Soil and Water Assessment Tool (SWAT) to simulate the hydrological cycle components in Xihe River Basin in northern China. The influence of climate change and land use change on green water resources in the basin from 1995 to 2015 is analyzed and compared using hypothetical scenarios. The paper is a typical application study but is quite useful for the study area and is generally well-structured as it explains the methodology, the mathematical framework and the assumptions used, and the justification of the results and the conclusions. However, there are few critical points that should be addressed in the revised manuscript. Addressing these comments will improve the quality of the paper and help the general reader of the paper.

1.      Climate change scenarios. Climate change is not analyzed in this study. Two historic periods are separated with no scientific evidence and used as climate change scenarios. My recommendation to the authors is to use data from several GCMs /RCMs to estimate future climate change scenarios. Why the meteorological data are separated in two periods (1990-1999 and 2006-2015) and how? Usually a statistical test is used for data separation in two (or more) periods. A table with the statistical characteristics of the two periods should be included in the revised manuscript for comparison purposes. Furthermore, I would like to see in the revised manuscript a statistical method for data separation. Then, the derived two periods could be used for model calibration and validation using the two land use scenarios. Please address these issues in the revised manuscript.

2.      Novelties of the method. Please discuss the novelties of the manuscript in comparison with previous studies at the study area. Please also highlight the similarities and differences of the previous works of the authors for the paper objectives.

3.      Land use validation. Please explain in details the procedure of the land use classification (classification algorithm and validation procedure). How the Google Earth is used in the land use classification method. A validation table for each individual class could exemplify the employed analysis. Please address this issue in the revised manuscript.  

Minor Comments

4.      Figure resolution. Please improve pixel resolution of Figures 1, 2, and 3. According to journal guidelines the figures should have high quality (>= 300 dpi).

5.      A flow diagram could be added in the revised manuscript for the international readers.

For the motivations listed above, the paper in its present form needs major revisions in order to evaluate the innovative character of the proposed method. The paper is of general interest for international audience and merits publication in Sustainability Journal when the major revisions and comments are addressed. Addressing these comments will improve the quality of the paper and help the general reader of the paper.

Author Response

Dear Reviewer:

Thank you for your comments concerning our manuscript entitled ‘sustainability-404523’. Those comments are all valuable and helpful for revising and improving our paper, the main corrections in the paper and the responds to your comments are as flowing:

1、We did have use statistical tests such as Mann-Kendall and Pettitt sudden change point detection method for meteorological data separation. However we didn’t find any sudden change point either in rainfall series or temperature series. So we chose 10 years (1990-1999 and 2006-2015) as our climate change scenarios, according the time of the land use data. The climate changes refer in particular to the differents between these two periods. [Zhao, Z. Z. Impact of Human Activities and Climate Variability on Green and Blue Water Resources in the Weihe River Basin of Northwest China. 2016, doi:10.13249/j.cnki.sgs.2016.04.011.]

2、There is no previous study on the spatial and temporal distribution of green water resources in the region in the semi-arid and semi-humid regions of northeast China. This study can provide some scientific advice for the management and application of water resources in northeast China.

3、We use the method of random sampling to verify the interpretation results. Firstly, multiple verification points are selected randomly. And then they are visually interpreted through original Google earth and TM images. We takes the visual interpretation results as reference truth values for human-computer interactive verification. And the accuracy requirements of remote sensing images interpreted in this paper are based on this references: [Janssen, L. L. F. , & Wel, F. J. M. V. D. . (1994). Accuracy assessment of satellite derived land cover data: a review. Photogrammetric Engineering & Remote Sensing, 60(4), 419-426. ]

4、We have improves the resolution of the figures.

5、We have added a flow diagram in the revised manuscript.

Round  2

Reviewer 2 Report

I would like to remind the first comment in the current effort "First of all, I would like to noted that, scenarios provide a mechanism for the forecasting, thus it is a tool of long-range outlook and it is related with various approaches including the water resources. In current effort, authors used the term for three deferent past periods as portrayed in table 1. However, we can say that scenario is defined as a or more future history / ies. I suggest to change the term or please justify in the introduction the term selection."  

Author Response

Dear Reviewer:

Thank you very much for your suggestion.

Response: We have added references for “Simulation Scenarios” according to your suggestions. (Ref. Senent-Aparicio, J.; Liu, S.; Pérez-Sánchez, J.; López-Ballesteros, A.; Jimeno-Sáez, P. Assessing Impacts of Climate Variability and Reforestation Activities on Water Resources in the Headwaters of the Segura River Basin (SE Spain). Sustainability 2018, 10, 3277.  And Zhao, A. Z., Zhu, X. F., Liu X.F., Pan, Y.Z. Zuo D.P., Impacts of land use change and climate variability on green and blue water resources in the Weihe River Basin of northwest China. Catena, 2016, 137: 318-327. ).

And:

1.We have add a validation table for land use in the revised manuscript.

2. We have renumbered the references of the manuscript.

Reviewer 3 Report

The paper in its present and revised form merits publication in MDPI Sustainability journal with minor revisions. The authors have addressed most of my major and minor comments, which are explained and analysed in the authors’ response documents. However, one major comment (comment No 1 of the 1st review) is not addressed in the revised manuscript. Based on this comment the “climate change” part of the manuscript is meaningless. Hence, I would recommend to the authors to use GCMs /RCMs to estimate future climate change scenarios if they want to address climate change. If not the term “climate change” should be removed from the title with subsequent changes in the text. Furthermore, the following revisions are recommended:

1. Comment 3 is partly addressed and I would like to see in the revised manuscript a validation table for each identified land use type.

2. Numbering of references. Please renumber the references in the revised manuscript. The addition of new references should follow the numbering in the text.

As a conclusion, based on the authors’ response to reviewers and the revised manuscript, I recommend minor revisions for the revised manuscript since most of the major and minor comments have been addressed.

Author Response

Dear Reviewer:

Thank you for your suggestion.

Response: We are quite in agreement with your suggestion. Scenarios can show the differences in climate, but not continuous climate change. So we have change the description from climate change to climate variability. (Ref. Senent-Aparicio, J.; Liu, S.; Pérez-Sánchez, J.; López-Ballesteros, A.; Jimeno-Sáez, P. Assessing Impacts of Climate Variability and Reforestation Activities on Water Resources in the Headwaters of the Segura River Basin (SE Spain). Sustainability 2018, 10, 3277.  And Ref. Yan, T.; Bai, J.; LEE ZHI YI, A.; Shen, Z. SWAT-Simulated Streamflow Responses to Climate Variability and Human Activities in the Miyun Reservoir Basin by Considering Streamflow Components. Sustainability 2018, 10, 941.).

And:

1.We have add a validation table for land use in the revised manuscript.

2. We have renumbered the references of the manuscript.
